# Characterization of Electrical Traps Formed in Al_2_O_3_ under Various ALD Conditions

**DOI:** 10.3390/ma13245809

**Published:** 2020-12-19

**Authors:** Md. Mamunur Rahman, Ki-Yong Shin, Tae-Woo Kim

**Affiliations:** School of Electrical Engineering, University of Ulsan, Ulsan 44610, Korea; rahman.mamun37@gmail.com (M.M.R.); rldyd3140@ulsan.ac.kr (K.-Y.S.)

**Keywords:** ALD, border trap, high-*k*, interface trap, III–V semiconductors

## Abstract

Frequency dispersion in the accumulation region seen in multifrequency capacitance–voltage characterization, which is believed to be caused mainly by border traps, is a concerning issue in present-day devices. Because these traps are a fundamental property of oxides, their formation is expected to be affected to some extent by the parameters of oxide growth caused by atomic layer deposition (ALD). In this study, the effects of variation in two ALD conditions, deposition temperature and purge time, on the formation of near-interfacial oxide traps in the Al_2_O_3_ dielectric are examined. In addition to the evaluation of these border traps, the most commonly examined electrical traps—i.e., interface traps—are also investigated along with the hysteresis, permittivity, reliability, and leakage current. The results reveal that a higher deposition temperature helps to minimize the formation of border traps and suppress leakage current but adversely affects the oxide/semiconductor interface and the permittivity of the deposited film. In contrast, a longer purge time provides a high-quality atomic-layer-deposited film which has fewer electrical traps and reasonable values of permittivity and breakdown voltage. These findings indicate that a moderate ALD temperature along with a sufficiently long purge time will provide an oxide film with fewer electrical traps, a reasonable permittivity, and a low leakage current.

## 1. Introduction

The aggressive scaling of the equivalent oxide thickness (EOT) of metal–oxide–semiconductor field-effect transistors (MOSFETs) for achieving sophisticated device speeds and minimal power consumption has made SiO_2_ obsolete as an insulating material [1,2,3]. As a result, oxide materials, which have a high dielectric constant (*k*), have emerged as the most suitable alternatives from the viewpoint of the realization of faster devices and overcoming the high leakage current problem [2,4,5]. Although many studies have reported several candidate high-*k* dielectric materials, Al_2_O_3_ has attracted attention as the most desirable of such materials, predominantly because of its comparatively larger band gap (which provides a better band alignment with the channel material), better thermal solidity, lower oxygen and ionic transportation, and more effective inactivation of interface traps in the channel material [5,6,7,8]. Besides the new insulating material alteration, the long-established Si is also out of contention as a channel material because of its limitations in the realization of faster devices; III–V compound semiconductors have instead been drawing attention for this purpose [6,9]. 

Among the various III–V semiconductors proposed as successor channel materials, In*_x_*Ga_1-*x*_As with *x* = 0.53 has proved to be rather promising because of its much higher electron mobility than Si (roughly eight times higher) and high injection velocity; these properties make it a suitable candidate for defense and high-frequency analog applications [9,10]. However, these advantages are offset by the disadvantage of its lower density of states, which is about two orders of magnitude lower than that of Si; because of this lower density of states, the Fermi level is pinned within the conduction band [10,11,12]. This pinning of the Fermi level causes the devaluation of the barrier height of high-*k*/In_0.53_Ga_0.47_. As with respect to the SiO_2_/Si one. Because of this reduction in band orientation elevation, the Fermi level is aligned with the energy level of near-interfacial oxide traps [9,13]. These traps are commonly known as border traps, which are an inherent property of oxides [13,14,15]. Because of this above-explained alignment of the Fermi level with the energy band of border traps, these traps capture or release the channel electrons via tunneling. This carrier exchange time is governed mainly by the applied AC frequency, where a lower frequency enables deeper traps to respond, and vice versa [15,16]. This frequency dependence, in turn, causes a discrepancy in the capacitance values, especially in the accumulation region. This tunneling also prevents the generation of a sufficient number of carriers, which results in low mobility and, in turn, leads to a decrease in the on-state current, transconductance, and reliability because of high hysteresis [15,16,17]. Moreover, it has been reported that when a specific voltage stress is applied to the device, electrons are more likely to become trapped in these oxide traps [18,19]. 

Various deposition approaches for high-*k* metal oxides have been reported thus far—e.g., sputtering, physical vapor deposition, chemical vapor deposition (CVD), and atomic layer deposition (ALD) [2,20,21,22]. Among such approaches, ALD has drawn considerable attention because its process is sophisticated, and it enables the precise control of the deposition thickness. The fundamental mechanism of film deposition by ALD is the chemical response of two vaporous reactants, commonly known as precursors, on the substrate surface, where the reactants are present, each in turn, in a successive non-covering way. This process is continued in such a way that the two precursors are never present together on the substrate surface at any given time; this process makes ALD unique vis-à-vis CVD, even though the former is a subclass of the latter [2]. Purge gas flow is introduced in between the application of two successive precursor pulses to remove the unreacted reactants, because the reactions stop by themselves (i.e., self-terminate) once all the reactive species on the surface are consumed. The combination of the application of one precursor pulse and a single introduction of purge gas flow is referred to as a half-cycle, and the amount of film deposited in two consecutive half-cycles is termed the growth per cycle (GPC). Therefore, the thickness of the deposited film can be easily controlled by varying the number of ALD cycles once the GPC is known; in contrast, in CVD the deposition thickness is maintained by particular time allocation. Thus, because of the self-terminating nature of the reaction and possibility of the precise control of thickness by ALD, it has recently been used quite extensively for depositing controlled, condensed, and pin hole-free high-quality film [20,23]. 

Several process parameters of ALD completely control film growth, one of which is the deposition temperature. It is well known that each precursor used in ALD has a temperature window in which it transforms into its reactive components. The temperature of the ALD chamber must be high enough to prevent the condensation of any of the reactive components and to consequently prevent the occurrence of any undesirable and uncontrollable reactions. Additionally, in some reactions, the activation energy needs to be exceeded by the temperature. Given these requisites, the temperature of the deposition chamber must be maintained at or above a certain minimum value. However, an excessively high deposition temperature may cause the reactant to decompose to an inappropriate extent, which would undesirably result in the occurrence of a CVD reaction. Furthermore, in some cases the re-evaporation of the deposited film occurs, which results in a lower GPC [24]. Therefore, the deposition temperature should be maintained within the temperature window to ensure efficient deposition by ALD. Another process parameter that greatly influences the efficiency of deposition by ALD is the purge time. The purge flow must meet the condition that the purge time should be just long enough to ensure the complete removal of all gas reactants and not any longer. If the unreacted gaseous components are not removed completely, CVD may possibly occur when the next reactant pulse is applied [25].

In terms of the electrical response of the border traps, they are rather different to the conventional interface traps. The frequency dispersion caused by border traps occurs mainly in the accumulation region, where interface traps are rather inactive. This border trap-induced frequency dispersion has a weak dependence on temperature. Additionally, this frequency dispersion is not affected by any chemical treatment, whereas the frequency dispersion caused by interface traps is reduced by chemical treatment [6,26]. Because the time taken by border traps to capture/emit an electron in the accumulation region is much longer than that taken by interface traps, conventional interface trap models are unable to characterize these oxide traps accurately [27,28]. In some previous works, border traps have been quantified using the capacitance–voltage (C–V) hysteresis, which lacks the complete re-emission of trapped charges when the C–V sweep is in a reverse direction [9]. Therefore, a characterization based on accumulation dispersion is more suitable for quantifying border traps. Furthermore, these oxide traps are believed to be formed at the time of oxide growth, which implies that the mechanism of ALD growth will have an impact on these traps. In this study, therefore, both of these kinds of electrical traps (border and interface traps) are characterized by varying two ALD conditions: the deposition temperature and the purge time. In addition, the stress responses of the films under constant voltage are also examined. 

## 2. Materials and Methods 

Al_2_O_3_ films were deposited at three different ALD temperatures (200, 250, and 300 °C) and with four different purge times (5, 10, 15, and 20 s). In our previous study, we deposited Al_2_O_3_ at a temperature of 250 °C with a purge time of 20 s and found that this condition provided the best GPC in our ALD system [6]. In the present study, we varied the deposition temperature by keeping the purge time fixed at 20 s and conversely varied the purge time by maintaining the deposition temperature at 250 °C. All these Al_2_O_3_ films were deposited on the *n*-In_0.53_Ga_0.47_As substrate, which was epitaxially grown on a typical 300 mm *n*-type Si (001) wafer, as described in our previous paper [6]. In the ALD process, trimethylaluminum (TMA) was used as the metal precursor, whereas H_2_O was used as the oxidant, and N_2_ was used as both the pulse and the purge gas with a flow rate of 300 sccm. Both the reactants were maintained at room temperature. 

Prior to ALD, the substrates were processed by standard cleaning procedures for the removal of contaminants and native oxide. They were first cleaned with acetone and isopropyl alcohol for 5 min each, after which they were cleaned with diluted HCl and deionized (DI) water (1:10 ratio) for 30 s at room temperature. In the final cleaning step, the wafers were cleaned with DI water for 2 min and dried in a N_2_ environment to prevent the formation of a water mask on the surface. The wafers were eventually transferred to the ALD chamber with minimal exposure to the external environment. In the ALD chamber, before the actual deposition the substrates were pretreated by being subjected to 10 cycles of precleaning with TMA because of its self-cleaning effect, which passivates the dangling bonds of the substrate and consequently minimizes the interface trap density [29,30]. The actual film deposition was commenced by the application of a TMA pulse to the substrate and the subsequent application of a water pulse. A purge flow was maintained between these two consecutive pulses. The pulse duration was 0.1 s in all cases. These four steps—i.e., the application of two consecutive pulses and the introduction of a purge flow after the application of each pulse—constituted a single ALD cycle, and this cycle was repeated 30 times for each case. 

The thickness of the deposited film samples was measured by ellipsometry at an elevation angle of 70°. The thicknesses of the films deposited at 200, 250, and 300 °C (purge time of 20 s) were 4.2, 3.9, and 3.5 nm, respectively, whereas those of the films deposited with purge times of 5, 10, and 15 s (deposition temperature of 250 °C) were 3.3, 3.4, and 3.8 nm, respectively. For the formation of metal–oxide–semiconductor capacitors (MOSCAPs), a 5 nm TiN metal layer was deposited by ALD on top of the Al_2_O_3_ dielectric film in all cases, which followed by deposition of a Ti/Au (200/2000 Å) layer via e-beam evaporation through a lift-off process for the realization of the front-side electrode. Another Ti/Au layer with a similar thickness was also deposited for the realization of the back-side contact. Reactive ion etching with SF_6_/Ar gas (30/10 sccm) was used to remove the TiN metal layer for the segregation of the MOSCAPs. For a reduction in the number of defects formed at the oxide/semiconductor and oxide/metal interfaces by metal deposition as well as film densification, with the eventual aim of decreasing the electrically active interface traps and border traps, all the devices were subjected to heat treatment in the form of rapid thermal annealing at 350 °C for 2 min in N_2_ ambient. The electrical characterization of the devices was performed using a Keithley 4200A-SCS parameter analyzer at room temperature and in a dark environment, and constant voltage stress (CVS) measurements were performed using a Keysight CV-enabled B1500A semiconductor device parameter analyzer. 

## 3. Results and Discussion

Figure 1 shows the measured multifrequency C–V responses for the two variation cases (variation in deposition temperature and variation in purge time) from 10 kHz to 1 MHz, along with the respective hysteresis curves. Figure 1a shows the measured responses for the three cases of deposition temperature variation. From the inversion responses obtained in the three cases, it is evident that the sample deposited at 300 °C has the lowest leakage current; it also has the lowest dispersion in the accumulation region, as depicted in Figure 1a, which indicates a lower density of border traps. Among the four samples deposited with different purge times, as depicted in Figure 1b, the sample with the 20 s purge time appears leakier than the others. Since the thickness of the deposited film varies from sample to sample, it is not possible to get a clear idea about the dielectric constant from the maximum value of the accumulation capacitance. Figure 1c,d show comparisons of the hysteresis curves for the two variation cases. The hysteresis was measured at a frequency of 1 MHz by starting the voltage sweep from inversion to accumulation and without any delay in sweeping back toward inversion. Among the samples deposited at different temperatures, as shown in Figure 1c, the 300 °C deposited sample shows the smallest hysteresis, which is another indication of a lower density of border traps. A larger hysteresis means that more charges are trapped into the near-interfacial oxide vacancies at the time when the Fermi level is in alignment with the trap energy level at accumulation condition. As a result of this, these captured charges will remain trapped in these traps until the Fermi level again comes closer to the valance band at the time of the reverse C–V sweep, which eventually causes a voltage shift. From the curves of the samples deposited with different purge times, it can be seen that the sample deposited with the longest purge time has the lowest hysteresis value, which implies that this film has the best stoichiometry. 

Figure 2 shows the calculated values of the effective dielectric constant (*k*_effective_) for the two variation cases (i.e., variation in deposition temperature and variation in purge time). The dielectric constant was calculated using the accumulation capacitance value at 10 kHz and the measured physical thickness obtained from the procedure described in our previous study. As shown in Figure 2a, the calculated *k*_effective_ values of the films deposited at 200, 250, and 300 °C are 6.93, 6.71, and 6.26, respectively. As shown in Figure 2b, the *k*_effective_ values of the films deposited with purge times of 5, 10, 15, and 20 s are 5.091, 5.23, 5.51, and 6.71, respectively. From these results, it is evident that a film deposited at a lower ALD temperature and with a longer purge time shows a higher permittivity, which means that it has better insulation properties. 

Figure 3 shows the fitted curves of the measured and calculated capacitances along with the plots of the border trap densities in the two variation cases. The measured capacitances were extracted from the accumulation capacitance in the multifrequency range (10 kHz–1 MHz) at the border trap extraction voltage. Since, at maximum bias voltage, charges fill the border traps that have a lower energy level than the Fermi level during CV hysteresis characterization, and at the reverse sweep hysteresis analysis the remission probability of these trapped charges from the border traps of a certain distance is quite low, the characterization of these traps by accumulation frequency dispersion around a solitary energy level at the maximum accumulation bias voltage (by assuming a spatial dispersal inside the oxide) is thus quite reasonable [28]. The capacitance used to characterize the border traps was calculated using the distributive border trap model proposed by Yuan et al., wherein a best fit condition was achieved at this calculated capacitance and measured capacitance by solving the following differential equation [31]:(1)$$\frac{dy}{dx}=-\frac{Y^{2}}{j\omega \varepsilon_{ox}}+\frac{q^{2}N_{bt}\ln(1+j\omega \tau )}{\tau }.$$

This equation has the boundary condition $Y=j\omega C_{s}$ at *x* = 0, where *Y* denotes the total admittance at any distance *x* from the oxide/semiconductor interface, *C_s_* is the semiconductor capacitance with a surface potential of ψ_s_, and *ω* is the angular frequency. In Equation (1), $\varepsilon_{ox}$ is the oxide capacitance, *N_bt_* is the volume concentration of border traps at a distance *x* inside the oxide, and *τ* is the average electron capture time. The effective electron mass is considered to be 0.23m_0_ for Al_2_O_3_ (where m_0_ denotes the electron rest mass), and *C_s_* is calculated using a one-dimensional Poisson–Schrodinger solver (Nextnano) at the border trap extraction voltage (which is 1 V in this case) [32,33]. In Figure 3a,b, some distortions are observed at lower frequencies in the measurement window, which may be caused by noise association at lower frequency measurements [6]. From the border trap density (*N_bt_*) plots in Figure 3c, it is observed that the number of oxide defects decreases with an increasing deposition temperature. The *N_bt_* values measured at the deposition temperatures of 200, 250, and 300 °C with a purge time of 20 s are 1.28 × 10^20^, 1.1 × 10^20^, and 1 × 10^20^ cm^−3^eV^−1^, respectively. The lowest *N_bt_* at 300 °C indicates that some stoichiometric changes may occur at higher deposition temperatures, which will, in turn, cause a decrease in the number of these traps at the time of oxide growth. The *N_bt_* values measured for purge times of 5, 10, 15, and 20 s at the deposition temperature of 250 °C are 1.18 × 10^20^, 1.23 × 10^20^, 1.48 × 10^20^, and 1.1 × 10^20^ cm^−3^eV^−1^, respectively. Although the *N_bt_* values fluctuate with the purge times, the lowest *N_bt_* value at the longest purge time implies the formation of a high-quality ALD film at this time. In contrast, the films deposited with shorter purge times may be defective because of the insufficient time available for unreacted reactants; that is, they may contain some residual reactants. Figure 4 shows plots of the interface trap density as a function of the deposition temperature and purge time as obtained by the conductance method—i.e., through the measurement of the parallel conductance (G_p_/ω_max_) with series resistance correction [34]. The interface trap density is then calculated as:(2)$$D_{it}=2.5\frac{\left(G_{p}/\omega_{\max}\right)}{Aq},$$ where *q* is the electron charge and A is the electrode area. From Figure 4a, it is observed that, although high-temperature deposition is better for decreasing the border traps, low-temperature deposition passivates the interface traps to some extent. Further, from Figure 4b, it is observed that, among the films deposited with various purge times, the film with a purge time of 20 s has the lowest *D_it_*, whereas that with a purge time of 10 s has the highest *D_it_*. 

The reliability of the deposited films under variation in the two deposition conditions was evaluated by CVS measurements at a 1 V bias for a duration of 1000 s, wherein the threshold voltage shift (V_TH_) was calculated by intersecting the stress after some explicit time border to consent the C–V quantification. The positive shifts in the threshold voltage in all the deposition cases indicate an increase in the trapping of negative charges in the oxide films. These positive voltage shifts may be attributed to electron trapping in the oxide and/or interface traps at the oxide/semiconductor interface and by assuming the magnitude of the shifts to be linearly proportional to the number of traps present in the films [35]. Because the lowest number of border traps is observed at the deposition temperature of 300 °C, the smallest voltage shift is also observed at this temperature, as shown in Figure 5a. However, an interesting result is observed for the films deposited with different purge times, as shown in Figure 5b: fewer charges are trapped in the films with purge times of 5 s and 10 s than in that with a purge time of 20 s, even though the former two films have numerous order traps. 

Figure 6 shows the measured current–voltage (J_G_–V) profiles as well as plots of the breakdown voltage and leakage current density for the two variation cases after the application of a positive bias voltage. The lowest leakage current (2.01 × 10^−9^ A/cm^2^) and highest breakdown voltage (6.5 MV/cm) of the film deposited at 300 °C may be attributed to the lowest number of oxide traps (as discussed earlier) in this film, which provide a path for the flow of leakage current. This trend is also observed for the films deposited at 200 and 250 °C. However, among the films deposited with various purge times, the film with a purge time of 20 s shows the lowest leakage current as well as the highest breakdown voltage, as is also observed from the above-described trap density plots, whereas the film with a purge time of 10 s shows a slightly lower leakage current despite having a higher trap density than that with a purge time of 5 s. 

## 4. Conclusions

In conclusion, the electrical traps formed in Al_2_O_3_ films deposited under variations in two ALD conditions (deposition temperature and purge time) are characterized and compared, given that the formation of oxide traps is related to some extent to film growth. The permittivity and interface trap density as well as the breakdown voltage of the three films deposited at different ALD temperatures under a fixed purge time show an increasing trend with increasing temperature, whereas the films’ hysteresis and border trap density show a decreasing trend with increasing temperature. The reliability and leakage current of these three films also show the latter trend because of the low quantity of traps extant. However, in the case of the films deposited with different purge times under a fixed deposition temperature, all of the abovementioned parameters are better at longer purge times, which indicates that the film stoichiometry is better at longer purge times because of the effective removal of the residual reactants. In summary, film deposition at a moderate temperature with a sufficiently longer purge time in the ALD process is effective in minimizing the formation of electrical traps. 

## Figures and Tables

**Figure 1 materials-13-05809-f001:**
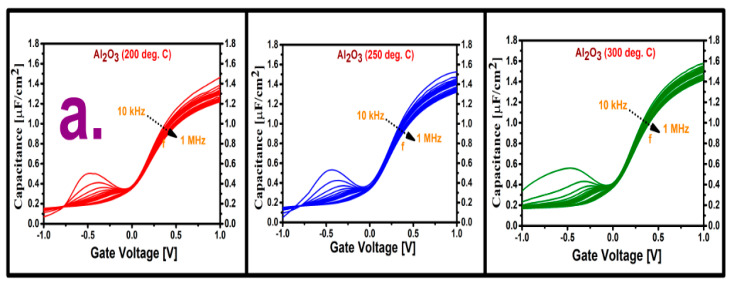
Frequency dispersion (10 kHz–1 MHz) C–V response of the annealed Al_2_O_3_ films (**a**) deposited at three temperatures (200, 250, and 300 °C) with a purge time of 20 s and (**b**) deposited with four purge times (5, 10, 15, and 20 s) at a deposition temperature of 250 °C. Comparison of hysteresis measured at 1 MHz from −1 V to +1 V for Al_2_O_3_ films (**c**) deposited at three temperatures (200, 250, and 300 °C) with purge time of 20 s and (**d**) deposited with four purge times (5, 10, 15, and 20 s) at a deposition temperature of 250 °C.

**Figure 2 materials-13-05809-f002:**
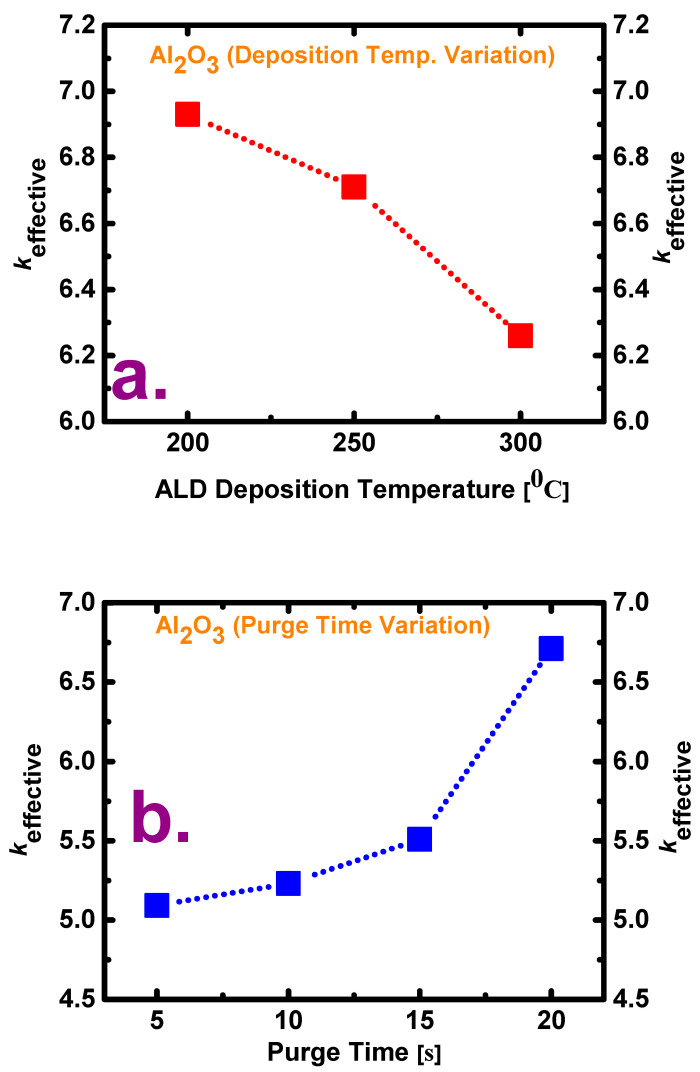
Comparison of the values of dielectric constant (*k*) for films (**a**) deposited at three temperatures (200, 250, and 300 °C) with a purge time of 20 s and (**b**) deposited with four purge times (5, 10, 15, and 20 s) at a deposition temperature of 250 °C.

**Figure 3 materials-13-05809-f003:**
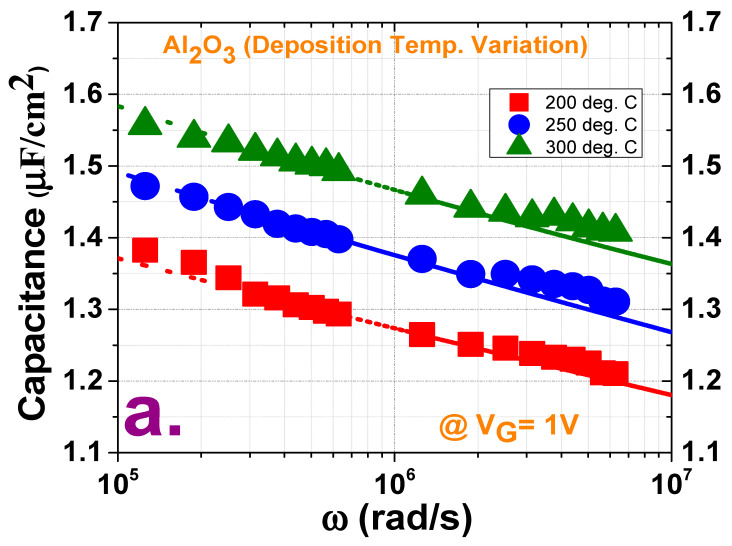
Best fitted curves of the measured capacitance values (symbols) and capacitance values calculated (dotted lines) for extracting the *N_bt_* using border trap model for films (**a**) deposited at three temperatures (200, 250, and 300 °C) with a purge time of 20 s and (**b**) deposited with four purge times (5, 10, 15, and 20 s) with a deposition temperature of 250 °C. Border trap densities calculated under variations in (**c**) deposition temperature and (**d**) purge time.

**Figure 4 materials-13-05809-f004:**
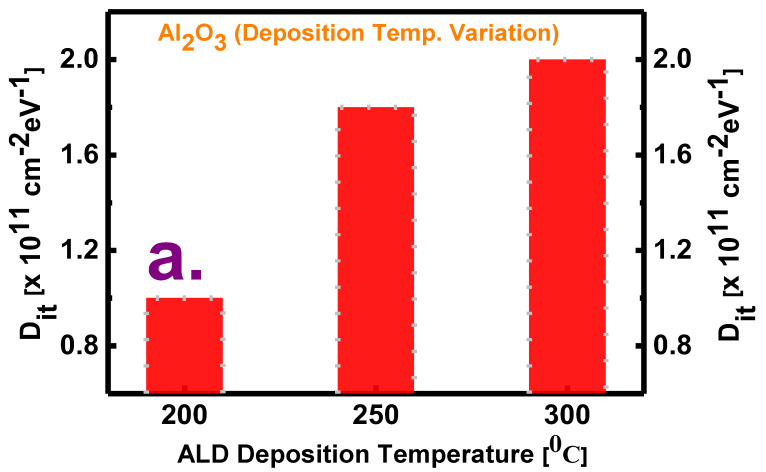
Comparison of the values of interface trap density (*D_it_*) measured for films (**a**) deposited at three temperatures (200, 250, and 300 °C) with a purge time of 20 s and (**b**) deposited with four purge times (5, 10, 15, and 20 s) at a deposition temperature of 250 °C.

**Figure 5 materials-13-05809-f005:**
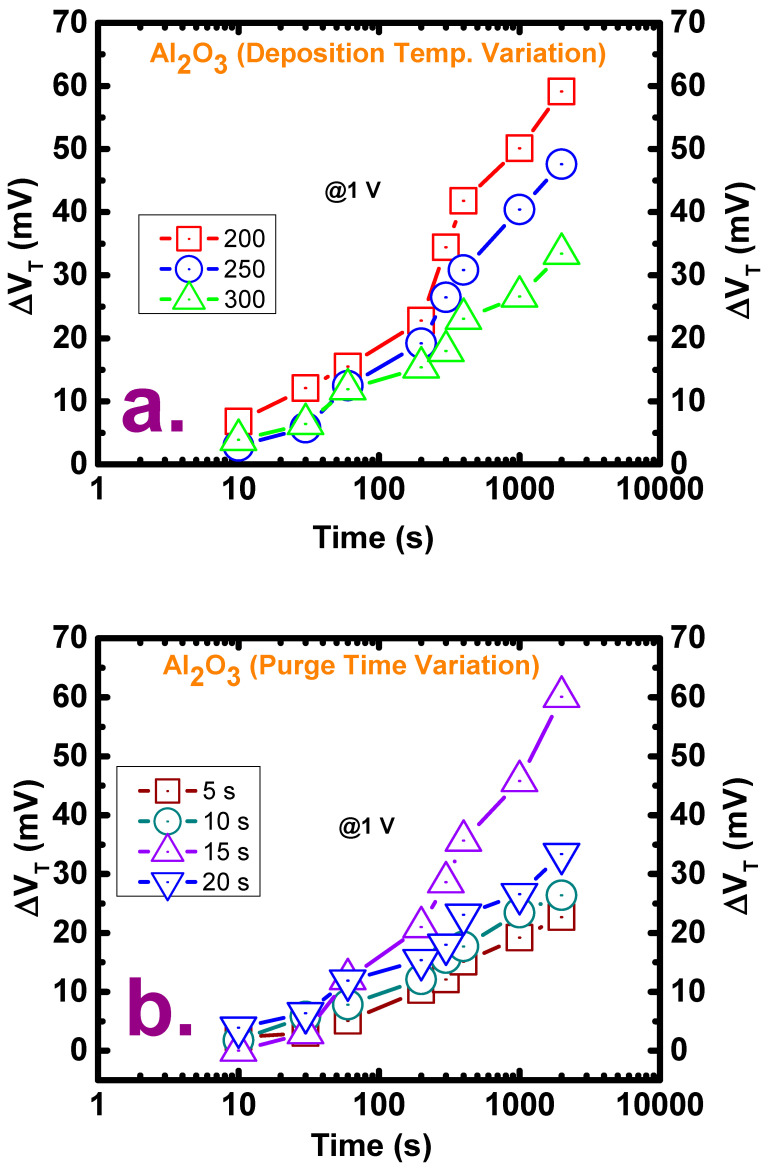
Threshold voltage shift (*V_TH_*) after the application of constant voltage stress (CVS) at a bias of 1 V for films (**a**) deposited at three temperatures (200, 250, and 300 °C) with a purge time of 20 s and (**b**) deposited with four purge times (5, 10, 15, and 20 s) at a deposition temperature of 250 °C.

**Figure 6 materials-13-05809-f006:**
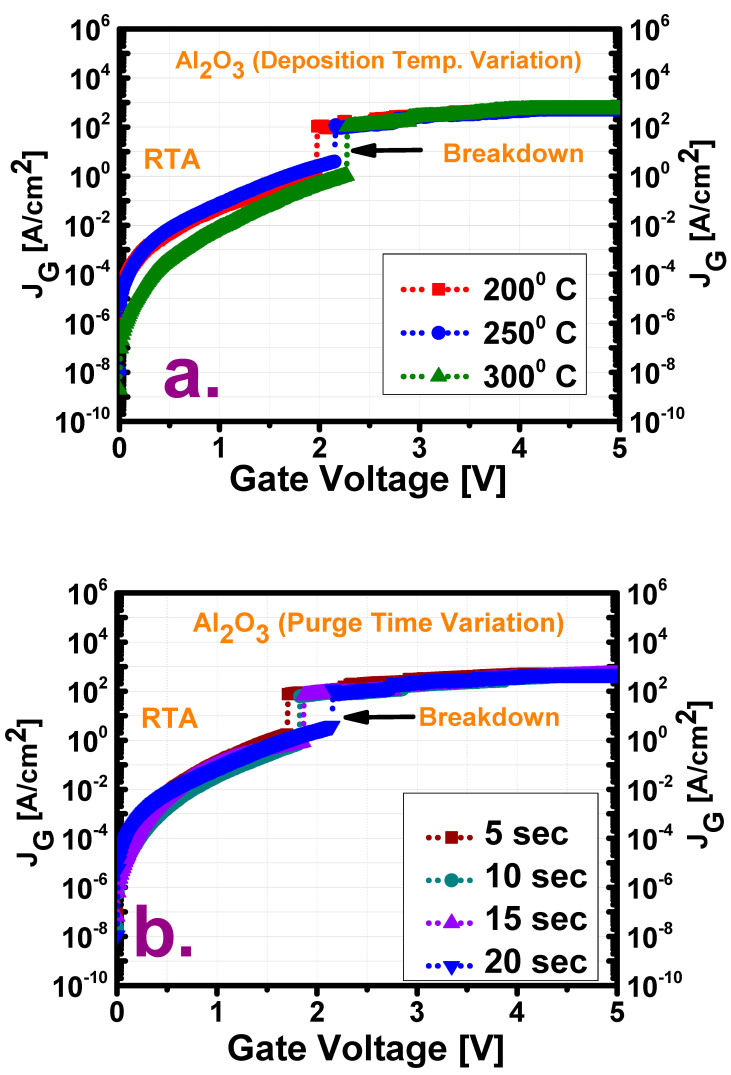
Leakage current–voltage (*J_G_*–*V*) profile under positive gate voltage for films (**a**) deposited at three temperatures (200, 250, and 300 °C) with a purge time of 20 s and (**b**) deposited with four purge times (5, 10, 15, and 20 s) at a deposition temperature of 250 °C. Comparison of the breakdown voltage (*V_BD_*) and leakage current density (*J_G_*) for films (**c**) deposited at three temperatures (200, 250, and 300 °C) with a purge time of 20 s and (**d**) deposited with four purge times (5, 10, 15, and 20 s) at a deposition temperature of 250 °C.

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
