# Peer review of "Characterization of Electrical Traps Formed in Al2O3 under Various ALD Conditions"

_materials, 2020, doi:10.3390/ma13245809_

Round 1

Reviewer 1 Report

Manuscript is devoted to study of correlation between electrical properties of Al2O3 thin film deposited by ALD and deposition condition. The films were examined in metal–oxide–semiconductor capacitors (MOSCAPs) configuration by multifrequency capacitance, constant voltage stress (CVS) and other electrical measurements. Obtained results was explained by presence of various type and density of electrical defects placed close to surface. It was found which deposition conditions (for actual ALD system) result in minimum of electrical trap density (in particularly border traps) and hence the best device properties.

The manuscript is well organized, experimental techniques are adequate chosen, results are mostly clearly described.

However, I have a few suggestions how the manuscript could be improved.

69 between the application of two successive precursor pluses to remove the unreacted reactants,

Type mistake

92 “The requisite for the purge flow is that the purge time should…” (English)

      The purge flow must meet the conditions that the purge time should…

140 “4.2006 nm, 3.867 nm, and 3.5128 nm, respectively, whereas those of the films deposited with purge  times of 5 s, 10 s, and 15 s (deposition temperature of 250 °C) were 3.2601 nm, 3.3507 nm, and 3.7818 142 nm, respectively”

In my opinion, the one digit in estimation of film thickness is more than enough. More digits have no physical sense.  

161 “From the inversion responses obtained in the three cases, it is evident that the sample deposited at 300 °C has the lowest leakage current; it also has the lowest dispersion in the accumulation region, …”

How it is seen?

171 “A larger hysteresis means that more charges are trapped into the near-interfacial  oxide vacancies at the time when at accumulation Fermi level is in alignment with the trap energy level and that they remain trapped in these traps until the Fermi level again comes closer to the valance band at the time of reverse C–V sweep, which eventually causes a voltage shift.”

The sentence is to long and unclear. It should be reformulated.

What is in Figs 3a and 3b? The Fig. caption should contain some additional explanation.

Author Response

Reviewer 1

Reviewer wrote:

Manuscript is devoted to study of correlation between electrical properties of Al2O3 thin film deposited by ALD and deposition condition. The films were examined in metal–oxide–semiconductor capacitors (MOSCAPs) configuration by multifrequency capacitance, constant voltage stress (CVS) and other electrical measurements. Obtained results was explained by presence of various type and density of electrical defects placed close to surface. It was found which deposition conditions (for actual ALD system) result in minimum of electrical trap density (in particularly border traps) and hence the best device properties.

The manuscript is well organized, experimental techniques are adequate chosen, results are mostly clearly described.

However, I have a few suggestions how the manuscript could be improved.

Our response:

            Dear the reviewer,

Thank you very much for carefully reviewing our manuscript and providing fruitful suggestions. We have taken all the comments into consideration, as below. We hope that the revision would be satisfactory to the reviewer and looking forward to hearing more comments.

Corresponding change in manuscript: No

Comment 1

Reviewer wrote:

69 - between the application of two successive precursor pluses to remove the unreacted reactants,

Type mistake

Our response:

Thanks for your valuable comment. We are really sorry for this typo. We have corrected it.

Corresponding change in manuscript: Typo is corrected.

Previous: between the application of two successive precursor pluses to remove the unreacted reactants,

New: between the application of two successive precursor pulses to remove the unreacted reactants

Location of change:

            Section 1: Introduction

            Page-2 and line 69.

Comment 2

Reviewer wrote:

92- “The requisite for the purge flow is that the purge time should…” (English)

      The purge flow must meet the conditions that the purge time should…

Our response:

Thank you for pointing out this issue. We are really sorry for this poor English. We have corrected as your instruction.

Corresponding change in manuscript: English has improved.

Previous: The requisite for the purge flow is that the purge time should…

New: The purge flow must meet the conditions that the purge time should be….

Location of change:

            Section 1: Introduction

            Page-2 and line 92.

Comment 3

Reviewer wrote:

140- “4.2006 nm, 3.867 nm, and 3.5128 nm, respectively, whereas those of the films deposited with purge times of 5 s, 10 s, and 15 s (deposition temperature of 250 °C) were 3.2601 nm, 3.3507 nm, and 3.7818 142 nm, respectively”

In my opinion, the one digit in estimation of film thickness is more than enough. More digits have no physical sense.

Our response:

Thank you for pointing this issue. We are really sorry for this inconvenience. We have made the necessary changes.

Corresponding change in manuscript: Only one digit is used to describe the film thickness.

Previous: The thicknesses of the films deposited at 200 °C, 250 °C, and 300 °C (purge time of 20 s) were 4.2006 nm, 3.867 nm, and 3.5128 nm, respectively, whereas those of the films deposited with purge times of 5 s, 10 s, and 15 s (deposition temperature of 250 °C) were 3.2601 nm, 3.3507 nm, and 3.7818 nm, respectively.

New: The thicknesses of the films deposited at 200 °C, 250 °C, and 300 °C (purge time of 20 s) were 4.2 nm, 3.9 nm, and 3.5 nm, respectively, whereas those of the films deposited with purge times of 5 s, 10 s, and 15 s (deposition temperature of 250 °C) were 3.3 nm, 3.4 nm, and 3.8 nm, respectively.

Location of change:

            Section 2: Materials and Methods

            Page-3 and line 139-141.

Comment 4

Reviewer wrote:

161 “From the inversion responses obtained in the three cases, it is evident that the sample deposited at 300°C has the lowest leakage current; it also has the lowest dispersion in the accumulation region, …”

How it is seen?

Our response:

Thank you for your comment. Actually, it is observed from Figure 1a. If we carefully consider the accumulation region of three cases, we can see that the 300°C sample has the lowest dispersion with respect to others, where a larger dispersion has observed at lower frequencies specially at 10kHz.  

Corresponding change in manuscript: The statement has slightly changed.

Previous: From the inversion responses obtained in the three cases, it is evident that the sample deposited at 300°C has the lowest leakage current; it also has the lowest dispersion in the accumulation region, which indicates a lower density of border traps.

New: From the inversion responses obtained in the three cases, it is evident that the sample deposited at 300°C has the lowest leakage current; it also has the lowest dispersion in the accumulation region as depicted in Figure 1a, which indicates a lower density of border traps.

Location of change:

            Section 3: Results and Discussion

            Page-4 and line 160-163.

Comment 5

Reviewer wrote:

171- “A larger hysteresis means that more charges are trapped into the near-interfacial oxide vacancies at the time when at accumulation Fermi level is in alignment with the trap energy level and that they remain trapped in these traps until the Fermi level again comes closer to the valance band at the time of reverse C–V sweep, which eventually causes a voltage shift.”

The sentence is too long and unclear. It should be reformulated.

Our response:

Thank you for your comment. We are really sorry for this inconvenience. We have broken it into two sentences and reformed it so that it will be clearer now. 

Corresponding change in manuscript: The sentence has parted into two sentences and reformed.

Previous: A larger hysteresis means that more charges are trapped into the near-interfacial oxide vacancies at the time when at accumulation Fermi level is in alignment with the trap energy level and that they remain trapped in these traps until the Fermi level again comes closer to the valance band at the time of reverse C–V sweep, which eventually causes a voltage shift.

New: A larger hysteresis means that more charges are trapped into the near-interfacial oxide vacancies at the time when the Fermi level is in alignment with the trap energy level at accumulation condition. And these captured charges will remain trapped in these traps until the Fermi level again comes closer to the valance band at the time of reverse C–V sweep, which eventually causes a voltage shift.

Location of change:

            Section 3: Results and Discussion

            Page-4 and line 171-175.

Comment 6

Reviewer wrote:

What is in Figs 3a and 3b? The Fig. caption should contain some additional explanation.

Our response:

Thank you for pointing this issue. We are really sorry for this inconvenience .Fig. 3a and 3b show the fitted curves of measured capacitance values (symbols) and capacitance values calculated (dotted lines) to calculate the border trap density using distributed border trap model for the two cases where Fig. 3a shows the various deposition conditions at three temperatures (200 °C, 250 °C, and 300°C) with purge time of 20 s and Fig. 3b shows the various deposition conditions at four purge times (5 s, 10 s, 15 s, and 20 s) with deposition temperature of 250 °C. We have edited the figure caption.

Corresponding change in manuscript: The figure caption is edited.

Previous: Fitted curves of measured capacitance values (symbols) and capacitance values calculated (dotted lines) using border trap model for films (a) deposited at three temperatures (200 °C, 250 °C, and 300°C) with purge time of 20 s and (b) deposited with four purge times (5 s, 10 s, 15 s, and 20 s) with deposition temperature of 250 °C.

New: Best fitted curves of measured capacitance values (symbols) and capacitance values calculated (dotted lines) for extracting the Nbt using border trap model for films (a) deposited at three temperatures (200 °C, 250 °C, and 300°C) with purge time of 20 s and (b) deposited with four purge times (5 s, 10 s, 15 s, and 20 s) with deposition temperature of 250 °C.

Location of change:

            Section 3: Results and Discussion

            Page-9 and line 209-213.

            Figure 3

Reviewer 2 Report

Comments to Authors

The manuscript entitled “Characterization of Electrical Traps Formed in Al2O3 under Various ALD Conditions” deals with the multifrequency C-V characterization of border and interface electrical traps formed in the Al2O3 by varying two ALD conditions: the deposition temperature and the purge time.

Although the metal/Al2O3/n-InGaAs device has been widely investigated as a substitute for conventional metal/SiO2/n-type Si field-effect-transistors (MOSFETs), limited works have been reported for the impact of Al2O3 ALD growth conditions on border and interface traps formation. Furthermore, it is the first time to my knowledge that an extensive work is present for the effects of variation of two ALD conditions, such as deposition temperature and purge time, on the formation of Al2O3/n-InGaAs interface electrical traps.

The experimental results presented in the manuscript are novel, original and well organized. In my opinion there are only few points needed revision:  

  1. According to fig.3d and 5b, the text lines 279-281 should be change to “fewer charges are trapped in the films with purge times of 5 s and 10 s than in that with the purge time of 20 s, even though the former two films have numerous border traps.”

  1. The fig.3d and 5b show that the film with the purge time of 20 s should have the smallest voltage shift because it has the lowest number of border traps. This is not seen in fig. 5b and an explanation must be given.

  1. In line 69, the word “pluses” should change to “pulses”.

  1. In line 229, the parameter, ξox, is not appeared in equation (1).

Based on the comments and minor revisions mentioned above, I propose to accept the manuscript.

Author Response

Reviewer 2

Reviewer wrote:

The manuscript entitled “Characterization of Electrical Traps Formed in Al2O3 under Various ALD Conditions” deals with the multifrequency C-V characterization of border and interface electrical traps formed in the Al2O3 by varying two ALD conditions: the deposition temperature and the purge time. Although the metal/Al2O3/n-InGaAs device has been widely investigated as a substitute for conventional metal/SiO2/n-type Si field-effect-transistors (MOSFETs), limited works have been reported for the impact of Al2O3 ALD growth conditions on border and interface traps formation. Furthermore, it is the first time to my knowledge that an extensive work is present for the effects of variation of two ALD conditions, such as deposition temperature and purge time, on the formation of Al2O3/n-InGaAs interface electrical traps. The experimental results presented in the manuscript are novel, original and well organized. In my opinion there are only few points needed revision:

Our response:

             Dear the reviewer,

Thank you very much for carefully reviewing our manuscript and providing fruitful suggestions. We have taken all the comments into consideration, as below. We hope that the revision would be satisfactory to the reviewer and looking forward to hearing more comments.

Corresponding change in manuscript: No

Comment 1

Reviewer wrote:

According to fig.3d and 5b, the text lines 279-281 should be change to “fewer charges are trapped in the films with purge times of 5 s and 10 s than in that with the purge time of 20 s, even though the former two films have numerous border traps.”

Our response:

Thanks for your comment. We are really sorry for our mistake. We have corrected it according to your suggestion.

Corresponding change in manuscript: The sentence has corrected.

Previous: fewer charges are trapped in the films with purge times of 10 s and 15 s than in that with the purge time of 20 s, even though the former two films have numerous border traps. 

New: fewer charges are trapped in the films with purge times of 5 s and 10 s than in that with the purge time of 20 s, even though the former two films have numerous border traps. 

Location of change:

            Section 3: Results and Discussion

            Page-13 and line 280-282.

Comment 2

Reviewer wrote:

The fig.3d and 5b show that the film with the purge time of 20 s should have the smallest voltage shift because it has the lowest number of border traps. This is not seen in fig. 5b and an explanation must be given.

  Our response:

Thank you for pointing out this issue. Yes, this result is also quite interesting to us that 5s and 10 s deposited films have lower captured charges as we mentioned in the manuscript. However, we suppose that these discontinuities have occurred due to some stoichiometric changes of the films at the time of device formation. However, further study is needed for a proper explanation which we are considering as our future plan.  

Corresponding change in manuscript: No.

Comment 3

Reviewer wrote:

In line 69, the word “pluses” should change to “pulses”.

Our response:

Thanks for your valuable comment. We are really sorry for this typo. We have corrected it.

Corresponding change in manuscript: Typo is corrected.

Previous: between the application of two successive precursor pluses to remove the unreacted reactants,

New: between the application of two successive precursor pulses to remove the unreacted reactants

Location of change:

            Section 1: Introduction

            Page-2 and line 69.

Comment 4

Reviewer wrote:

In line 229, the parameter, ξox, is not appeared in equation (1).

Our response:

Thank you for pointing this issue. We are really sorry for this inconvenience. We have corrected the proper symbol.

Corresponding change in manuscript: Symbol has changed.

Previous: ξox

New:

Location of change:

            Section 3: Results and Discussions

            Page-10 line 230.